# Progressive exercise versus best practice advice for adults aged 50 years or over after ankle fracture: the AFTER pilot randomised controlled trial

David J Keene ![ORCID],[1,2] Matthew L Costa,[1] Nicholas Peckham,[3] Elizabeth Tutton ![ORCID],[1,4] Vicki S Barber,[3] Susan J Dutton,[3] Sally Hopewell,[3] Anthony C Redmond,[5] Keith Willett,[1] Sarah E Lamb,[2,3] on behalf of the AFTER pilot trial collabororators

For numbered affiliations see end of article.

**Correspondence to**
Dr David J Keene;
david.keene@ndorms.ox.ac.uk

## ABSTRACT

**Objective** The aim of the Ankle Fracture Treatment: Enhancing Rehabilitation (AFTER) study, a multicentre external pilot parallel-group randomised controlled trial (RCT), was to assess feasibility of a definitive trial comparing rehabilitation approaches after ankle fracture.

**Setting** Five UK National Health Service hospitals.

**Participants** Participants were aged 50 years and over with an ankle fracture requiring immobilisation for at least 4 weeks.

**Interventions** Participants were allocated 1:1 via a central web-based randomisation system to: (1) best practice advice (one session of physiotherapy, up to two optional additional advice sessions) or (2) progressive exercise (up to six sessions of physiotherapy).

**Primary outcome measures** Feasibility: (1) participation rate, (2) intervention adherence and (3) retention.

**Results** Sixty-one of 112 (54%) eligible participants participated, exceeding progression criteria for participation of 25%. Recruitment progression criteria was 1.5 participants per site per month and 1.4 was observed. At least one intervention session was delivered for 28/30 (93%) of best practice advice and 28/31 (90%) of progressive exercise participants, exceeding the 85% progression criteria. For those providing follow-up data, the proportion of participants reporting performance of home exercises in the best practice advice and the progressive exercise groups at 3 months was 20/23 (87%) and 21/25 (84%), respectively. Mean time from injury to starting physiotherapy was 74.1 days (95% CI 53.9 to 94.1 days) for the best practice advice and 72.7 days (95% CI 54.7 to 88.9) for the progressive exercise group. Follow-up rate (6-month Olerud and Molander Ankle Score) was 28/30 (93%) for the best practice advice group and 26/31 (84%) in the progressive exercise group with an overall follow-up rate of 89%.

**Conclusions** This pilot RCT demonstrated that a definitive trial would be feasible. The main issues to address for a definitive trial are intervention modifications to enable earlier provision of rehabilitation and ensuring similar rates of follow-up in each group.

**Trial registration number** ISRCTN16612336.

## STRENGTHS AND LIMITATIONS OF THIS STUDY

⇒ Our pilot randomised controlled trial demonstrated that a definitive trial comparing a best practice advice versus a supervised progressive exercise intervention for adults aged 50 years and over after ankle fracture is feasible.

⇒ All feasibility success criteria were met, apart from a minor difference in the recruitment rate per site-month that would need to be factored into planning a definitive trial.

⇒ Follow-up rates differed between the groups; therefore, ensuring similar rates in a definitive trial is critical.

⇒ A limitation of the study, common to all pilot trials, is that with the necessarily limited number of centres, there remains some uncertainty regarding the feasibility of a trial across a much larger number of centres.

## INTRODUCTION

Ankle fractures account for 9% of all fractures managed in secondary care.[1] In the UK, incidence of these fractures is highest in people aged 50 years and over, peaking at 16 per 10 000 person-years in women aged 60–70.[2] As the population ages, a threefold increase in these fractures is projected over the next two decades.[3] The mechanism of injury for people aged over 50 is commonly a fall from standing height, which are defined as fragility fractures.[4]

Treatments for ankle fractures range from conservative plaster casts or boots to surgical fixation. The ankle injury management trial found that regardless of the initial fracture management, at 6 months reduced ankle function and walking abnormalities are substantial.[5 6] Participants reported an average 30% loss of preinjury ankle function. Function is poor due to

**Table 1** Intervention descriptions based on TIDieR guidance

| TIDieR items[35] | Description | |
|---|---|---|
| Brief name | AFTER (Ankle Fracture Treatment: Enhancing Rehabilitation) | |
| Why | Physiotherapy-led exercise and advice are commonly used to supervise rehabilitation after ankle fracture, but evidence is lacking in terms of its benefit over advice on self-management. | |
| | Progressive exercise | Best practice advice |
| What | Home exercise and advice programme overseen by physiotherapist over ≤6 sessions within 16 weeks. | Home exercise and advice programme initiated during a single face-to-face session with a physiotherapist, and then performed unsupervised by participant at home. |
| Materials: participants | Participant information booklet and exercise instruction sheets with photos. Action planner and exercise diary. Resistance bands (if applicable). | Participant information booklet incorporating exercise instructions with photos. |
| Materials: physiotherapists | Therapist training materials: training session pack detailing study and intervention procedures and a quick reference guide detailing all aspects of the progressive exercise intervention. | Therapist training materials: training session pack detailing study and intervention procedures. |
| Training | Up to 4 hours face-to-face training delivered by AFTER trial research physiotherapist. Training pack detailing all aspects of the trial and both interventions. | |
| Procedures | Initial appointment: Assess participant as per normal physiotherapy practice. Issue folder containing the progressive exercise participant information booklet. Agree level of exercise that is most appropriate for the participant initially. Advice should address barriers to exercise identified during assessment. Provide education regarding pain during and after exercise. Help participant to complete exercise documentation (exercise diary and action planner). Make follow-up appointment(s). Complete treatment log.  Appointments 2–6: Reassess as per normal physiotherapy practice. Assist participant to progress/regress exercises. Reassure the participant, reinforce key messages from the advice and education. Review home exercise programme using the exercise diary. Discuss return to functional activities. Review action planner. Complete treatment log (after every session). | Single face-to-face appointment: Assess participant as per normal physiotherapy practice. Issue best-practice advice participant information booklet. Explain exercise difficulty level to start on based on assessment. Educate participants how to progress and regress their exercises. Advice should address barriers to exercise identified during assessment. Provide education regarding pain during and after exercise. Discharge participant with advice/encouragement to continue with the self-management exercise programme for at least 16 weeks. Complete treatment log.  Up to two further advice sessions (optional for participants having difficulties with self-management or the exercises). Physiotherapist's role was to re-assess and re-enforce self-management advice. |
| Who provides | Physiotherapists already working in NHS musculoskeletal services who have attended the progressive exercise intervention training. AFTER did not exclude physiotherapists based on number of years qualified or experience. | Physiotherapists already working in NHS musculoskeletal services who have attended the best practice advice intervention training. AFTER did not exclude physiotherapists based on number of years qualified or experience. |
| How | Participants receive up to six sessions with a physiotherapist over 16 weeks. The initial session 20–60 min for assessment. It is then followed by up to five sessions of up to 30 min each. | Participants receive a single face-to-face session with a physiotherapist lasting 20–60 min. Up to two further advice sessions were optional. |
| Where | Physiotherapy sessions were in outpatient clinics based in the UK NHS. Exercise programme is performed by the participant at home. | Same as progressive exercise intervention. |
| When and how much | Initial appointment was as soon as possible after splint/cast removal, as per local appointment availability. Up to five follow-up sessions arranged within 16 weeks. Could be less than six sessions if participant has met rehabilitation goals and was self-managing condition. | The initial appointment was the same as progressive exercise in terms of timing, but additional one or two sessions were optional for participants that were having difficulties with self-management or the exercises. |
| Tailoring | Education and advice: Focus of education and advice were individualised based on assessment.  Exercises: Selection, manipulation of sets, repetitions and/or load is a joint decision-making process. Range of motion and position could be modified to accommodate the patient's comfort and preferences. | Education and advice: Focus of education and advice are individualised based on assessment.  Exercises: The range of motion through which an exercise is performed, and the load and volume, could be increased or decreased. |
| Modifications | The optional follow-up sessions for best practice advice were initially set as face-to-face or telephone. Later in recruitment this was altered to telephone only. | |
| Intervention fidelity | | |

Continued

**Table 1** Continued

| TIDieR items[35] | Description | |
|---|---|---|
| How well: training | All aspects of training delivery, content, structure, duration and therapists' confidence to implement the intervention were evaluated using post training feedback forms completed anonymously. | |
| How well: physiotherapists | Intervention fidelity was monitored centrally via treatment logs, and during site visits. | |
| How well: participants | Exercise adherence: Physiotherapists review the exercise diary at each subsequent session. Participants asked to report exercise frequency in postal follow-up questionnaires at 3 and 6 months. | Exercise adherence: Participants asked to report exercise frequency in postal follow-up questionnaires at 3 and 6 months. |
| How well: reporting | Intervention delivery data captured on treatment logs was entered onto the trial database and the findings are reported in the results section of this manuscript. | |

NHS, National Health Service; TIDieR, Template for Intervention Description and Replication .

pain and reduced joint motion,[7] lower limb muscle strength deficits,[8] gait abnormalities[9] and resultant mobility limitations.[7 10]

Weight bearing and ankle movement restrictions are usually removed by the orthopaedic team 6 weeks after injury. A Cochrane review[11] of ankle fracture rehabilitation concluded that there was insufficient evidence to support traditional physiotherapy interventions targeting ankle joint and muscle impairments, such as stretching,[12] manual therapy[13] and exercise.[14] Updating the Cochrane review searches in MEDLINE and Embase, identified another multicentre trial by Moseley and colleagues.[15] They found no differences in self-reported lower limb function or quality of life between supervised exercise and a one-off advice session for adults with ankle fractures treated surgically and conservatively. Physiotherapists delivered both interventions face-to-face and the mean age of participants was 42 years.

Physiotherapy interventions tested to date have not incorporated features of advice and exercise programmes for older adults used in other rehabilitation areas. Issues such as persistent poor balance in older adults require attention.[16] The Ankle Fracture Treatment: Enhancing Rehabilitation (AFTER) study was a pilot randomised controlled trial (RCT) to assess feasibility of a definitive trial comparing best practice advice with a supervised progressive exercise programme for adults aged 50 years and over after ankle fracture.

### Feasibility objectives
The main objectives of the pilot trial were to:
► Assess patient engagement with the trial, measured by the participation rate of those eligible.
► Establish whether the interventions were acceptable to participants and therapists, measured by intervention adherence levels (and participant interviews and a therapist focus group—to be reported separately).
► Determine patient retention, measured by the proportion of patients providing outcome data at 6 months.
► Assess the acceptability of measuring outcomes at 3 and 6 months post randomisation, measured by outcome measure data collection.

## METHODS
### Study design
A multicentre pilot parallel-group RCT. Participants were allocated to either: (1) best practice advice (one session of face-to-face advice delivered by a physiotherapist, with up to two further optional advice sessions as deemed required by the physiotherapist) or (2) progressive exercise (up to six sessions of individual face-to-face physiotherapy). The pilot trial protocol has been published.[17] An embedded qualitative study including patient participants and staff was also completed and will be reported separately.

### Setting
Recruitment was from five National Health Service (NHS) hospitals. Participants were identified while inpatients or when attending fracture clinics, where they were screened and given study information. Patients who meet the eligibility criteria and wanted to participate were approached for written informed consent.

### Study participants
The target population was adults aged 50 years as bone density reduces at this age and there is an increased risk of fragility fractures.[18] There is also a bimodal age distribution for ankle fractures, with peak incidences being for young adult men and highest for women aged 50 years and over.[2] Participants were attending NHS services to manage ankle fractures that either required definitive management with surgical, or non-surgical management of ankle immobilisation for at least 4 weeks.

Patients were excluded if they:
► Were unable to adhere to trial procedures or complete questionnaires.
► Did not have capacity to consent to study participation.
► Were not ambulatory before the injury.
► Were considered inappropriate for referral to physiotherapy by the clinician.
► Could not attend outpatient physiotherapy at a participating centre.
► Had serious concomitant disease (such as terminal illness).

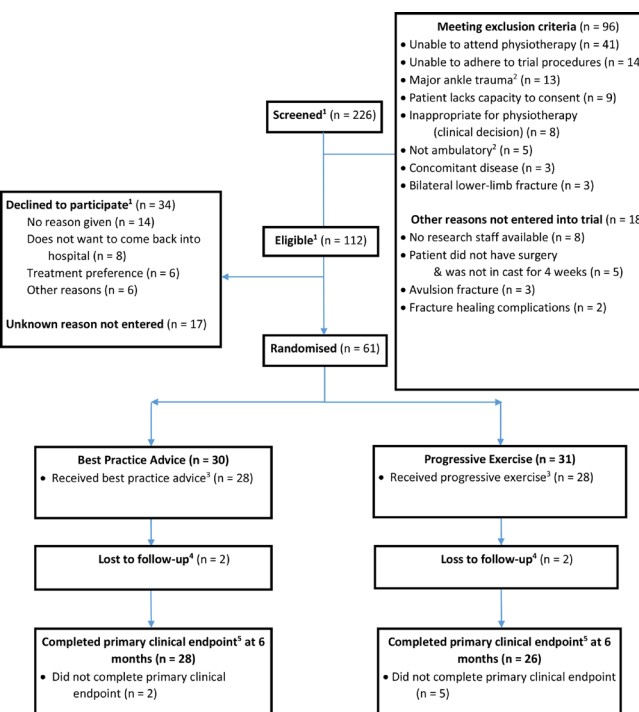

**Figure 1** Flow of participants through the Ankle Fracture Treatment: Enhancing Rehabilitation pilot trial. [1]One site has missing screening data for some of the randomised participants, so those who were randomised were added to the totals screened. [2]Major ankle trauma defined as: open fracture wounds, external fixation or substantial skin loss or grafts, that would limit ankle or lower leg exercise. [3]Compliance is defined as attending at least one session. [4]Loss to follow-up is defined as no response at the 6-month follow-up. [5]Olerud and Molander Ankle Score scores at 6 months.

► Had bilateral lower limb fractures.
► Had an ipsilateral concurrent Pilon fracture.
► Had open fracture wounds, external fixation or substantial skin loss or grafts that would limit ankle or lower leg exercise.

## Randomisation

Participants were randomised 1:1 to the intervention groups using the centralised computer randomisation service provided by the Oxford Clinical Trials Research Unit. The site's research facilitator undertook randomisation directly themselves or contacted the study office over the telephone to access the system on their behalf. Randomisation was computer-generated and stratified by centre and initial fracture management (surgery or non-surgical) using variable block sizes (2 and 4).

## Blinding

Physiotherapists delivering the intervention and study participants were told the treatment allocation. Physiotherapists or trained research associates independent of the clinical team collected the objective outcome measures at the 6-month follow-up time point.

## Interventions

The best practice advice or progressive exercise sessions were given when the participant could mobilise with unrestricted weight-bearing and do ankle exercises as guided by their surgeon or physiotherapist. We anticipated that this would usually be around 6–8 weeks post injury. Both interventions were delivered face-to-face and one-to-one by a physiotherapist. The interventions are outlined in table 1. In summary:

### Best practice advice

The best practice advice intervention included a 20–60 min session (depending on local service provision) of assessment, education, reassurance and detailed self-management advice on ankle exercises, gait training, stair climbing, walking aid advice and basic balance exercises. An advice booklet provided a summary of key information.

Up to two further advice sessions were optional for participants having difficulties with self-management or the exercises. The physiotherapist's role was to reassess and re-enforce self-management advice. During the initial stage of recruitment these additional sessions involved a telephone call or an additional face-to-face contact. In the latter stage of recruitment (after 41 participants had been recruited), these sessions were to be offered as telephone consultations only. This change was introduced to assess whether a greater focus on the single face-to-face session was acceptable to trial participants and physiotherapists. Use of additional sessions were recorded and monitored.

### Progressive exercise

A physiotherapist instructed participants and supervised their progressive exercise programme in up to six sessions over 16 weeks. This period aimed to allow sufficient time for neuromuscular adaptation to exercise.[19] The programme could end early if all rehabilitation goals were achieved in under six sessions. The first session was 20–60 min and the rest were up to 30 min, consistent with the duration of physiotherapy sessions in the NHS. Physiotherapists provided assessment, advice, education about progressing recovery, gait training, walking aid instruction and an information booklet.

The programme was highly structured but calibrated for each individual. All participants received a core set of functional lower limb resistance exercises in line with the evidence for improving muscular strength and power in older persons' rehabilitation.[20] The programme included supervised gait training to target major walking difficulties after ankle fracture. Balance exercises were included in the programme and introduced once the participant was able to weight bear sufficiently to perform these. Exercises were practised in the clinic but conducted at home with the aim of achieving an effective dose. Based on the participant's functional goals, exercises were progressed to make them task-specific, for example, walking on uneven surfaces or slopes, climbing stairs or jogging. Participants received an exercise planner

**Table 2** Baseline characteristics

| | Best practice advice | Progressive exercise | Total |
|---|---|---|---|
| Age (mean (SD)) | 63.3 (7.2) | 63.7 (10.4) | 63.5 (8.9) |
| Sex (n (%)) | | | |
| Female | 22 (73.3) | 20 (64.5) | 42 (68.9) |
| Male | 8 (26.7) | 11 (35.5) | 19 (31.1) |
| BMI* | 27.9 (4.8) | 29.0 (6.8) | 28.5 (5.9) |
| Ethnicity (n (%)) | | | |
| Black or black British | 1 (3.3) | 2 (6.5) | 3 (4.9) |
| Mixed | 1 (3.3) | 1 (3.2) | 2 (3.3) |
| Other | 1 (3.3) | 0 (0) | 1 (1.6) |
| White British | 27 (90.0) | 26 (83.9) | 53 (86.9) |
| White other | 0 (0) | 2 (6.5) | 2 (3.3) |
| Education (n (%)) | | | |
| Higher professional or university education | 17 (56.7) | 14 (45.2) | 31 (50.8) |
| None or primary education | 3 (10.0) | 0 (0) | 3 (4.9) |
| Secondary education | 10 (33.3) | 17 (54.8) | 27 (44.3) |
| Smoking status (n (%)) | | | |
| Current smoker | 3 (10.0) | 4 (12.9) | 7 (11.5) |
| Former smoker | 10 (33.3) | 15 (48.4) | 25 (41.0) |
| Never smoked | 17 (56.7) | 12 (38.7) | 29 (47.5) |
| If current smoker, number of cigarettes per day (mean (SD)) | 9.3 (6.9) | 17.5 (17.9) | 14.2 (14.9) |
| Alcohol units per week (mean (SD)) | 11.1 (13.4) | 5.5 (9.1) | 8.2 (11.6) |
| Employment status (n (%)) | | | |
| Employed | 11 (36.7) | 14 (45.2) | 25 (41.0) |
| Other | 1 (3.3) | 0 (0) | 1 (1.6) |
| Permanently sick or disabled | 0 (0) | 1 (3.2) | 1 (1.6) |
| Retired | 14 (46.7) | 14 (45.2) | 28 (45.9) |
| Semi retired | 4 (13.3) | 1 (3.2) | 5 (8.2) |
| Unemployed | 0 (0) | 1 (3.2) | 1 (1.6) |
| Time off work (due to injury) (n (%)) | | | |
| Not in paid work | 13 (43.3) | 15 (48.4) | 28 (45.9) |
| No | 4 (13.3) | 3 (9.7) | 7 (11.5) |
| Yes | 12 (40.0) | 12 (38.7) | 24 (39.3) |
| Work days missed (mean (SD)) | 16.1 (28.4) | 19.3 (19.1) | 17.9 (23.2) |
| Accommodation type (n (%)) | | | |
| House/flat rented from housing association/local authority | 3 (10.0) | 3 (9.7) | 6 (9.8) |
| Other | 2 (6.7) | 0 (0) | 2 (3.3) |
| Owner occupied house/flat | 21 (70.0) | 21 (67.7) | 42 (68.9) |
| Privately rented house/flat | 3 (10.0) | 7 (22.6) | 10 (16.4) |
| Sheltered housing/warden control | 1 (3.3) | 0 (0) | 1 (1.6) |
| Unpaid carer (n (%)) | | | |
| Yes | 5 (16.7) | 9 (29.0) | 14 (23.0) |
| No | 25 (83.3) | 22 (71.0) | 47 (77.0) |
| Resident unpaid carer (n (%)) | | | |
| Yes | 4 (13.3) | 5 (16.1) | 9 (14.8) |
| No | 1 (3.3) | 4 (12.9) | 5 (8.2) |
| No unpaid carer | 25 (83.3) | 22 (71.0) | 47 (77.0) |
| Paid carer (n (%)) | | | |
| Yes | 2 (6.7) | 2 (6.5) | 4 (6.6) |

**Table 2** Continued

| | Best practice advice | Progressive exercise | Total |
|---|---|---|---|
| No | 28 (93.3) | 28 (90.3) | 56 (91.8) |
| Unknown/missing | 0 (0) | 1 (3.2) | 1 (1.6) |
| Resident paid carer (n (%)) | | | |
| Yes | 1 (3.3) | 2 (6.5) | 3 (4.9) |
| No | 1 (3.3) | 0 (0) | 1 (1.6) |
| No paid carer | 28 (93.3) | 28 (90.3) | 56 (91.8) |
| Unknown/missing | 0 (0) | 1 (3.2) | 1 (1.6) |
| Treatment type (n (%)) | | | |
| Conservative (non-surgical) management | 15 (50.0) | 13 (41.9) | 28 (45.9) |
| Traditional cast non-removable | 3 (10.0) | 0 (0) | 3 (4.9) |
| Traditional cast removable | 1 (3.3) | 3 (9.7) | 4 (6.6) |
| Close contact casting | 5 (16.7) | 4 (12.9) | 9 (14.8) |
| Walking boot | 6 (20.0) | 6 (19.4) | 12 (19.7) |
| Unknown/missing | 3 (10.0) | 3 (9.7) | 6 (9.8) |
| Surgical management | 12 (40.0) | 15 (48.4) | 27 (44.3) |
| Clinical outcomes (mean (SD)) | | | |
| OMAS† | 31.38 (17.97) | 21.50 (15.71) | 26.36 (17.44) |
| LEFS‡ | 25.07 (13.39) | 17.52 (11.68) | 21.10 (12.98) |
| FES-I§ | 14.75 (5.18) | 17.00 (6.16) | 15.85 (5.74) |
| EQ-5D utility¶ | 0.48 (0.29) | 0.25 (0.30) | 0.36 (0.31) |
| EQ-5D VAS** | 69.52 (19.11) | 68.63 (20.04) | 69.07 (19.42) |
| Pain at rest VAS†† | 26.6 (28.2) | 32.1 (27.5) | 29.4 (27.7) |
| Pain walking VAS†† | 37.8 (30.0) | 55.2 (27.0) | 46.2 (29.6) |
| Falls (last 3 months) | 14.8 (5.2) | 17.0 (6.2) | 15.9 (5.7) |
| Falls that resulted in a broken bone (last 3 months) (n (%)) | | | |
| No | 12 (40.0) | 10 (32.3) | 22 (36.1) |
| Yes‡‡ | 2 (6.7) | 3 (9.7) | 5 (8.2) |
| Unknown/missing | 16 (53.3) | 18 (58.1) | 34 (55.7) |
| Aids preinjury (n (%)) | | | |
| Frame/rollator | 8 (26.7) | 1 (3.2) | 2 (3.3) |
| None | 5 (16.7) | 27 (87.1) | 51 (83.6) |
| One crutch | 3 (10.0) | 1 (3.2) | 1 (1.6) |
| One stick | 3 (10.0) | 0 (0) | 3 (4.9) |
| Two crutches | 8 (26.7) | 1 (3.2) | 1 (1.6) |
| Two sticks | 1 (3.3) | 1 (3.2) | 1 (1.6) |
| Unknown/missing | 2 (6.7) | 0 (0) | 2 (3.3) |
| Aids since injury (n (%)) | | | |
| Frame/rollator | 8 (26.7) | 13 (41.9) | 21 (34.4) |
| None | 5 (16.7) | 1 (3.2) | 6 (9.8) |
| One crutch | 3 (10.0) | 1 (3.2) | 4 (6.6) |
| One stick | 3 (10.0) | 0 (0) | 3 (4.9) |
| Two crutches | 8 (26.7) | 11 (35.5) | 19 (31.1) |
| Two sticks | 1 (3.3) | 2 (6.5) | 3 (4.9) |
| Unknown/missing | 2 (6.7) | 3 (9.7) | 5 (8.2) |
| Walking distance preinjury (n (%)) | | | |
| About the house | 1 (3.3) | 2 (6.5) | 3 (4.9) |
| Less than half a mile | 3 (10.0) | 4 (12.9) | 7 (11.5) |
| More than half a mile | 24 (80.0) | 25 (80.6) | 49 (80.3) |

Continued

**Table 2** Continued

|  | Best practice advice | Progressive exercise | Total |
|---|---|---|---|
| Bedbound | 0 (0) | 0 (0) | 0 (0) |
| Unknown/missing | 2 (6.7) | 0 (0) | 2 (3.3) |
| Medication use (n (%)) | | | |
| Yes | 21 (70.0) | 27 (87.1) | 48 (78.7) |
| No | 9 (30.0) | 4 (12.9) | 13 (21.3) |

*BMI: body mass index calculated as height divided by weight squared $(m/kg^2)$.
†OMAS: Olerud and Molander Ankle Score (0–100, higher scores better).
‡LEFS: Lower Extremity Functional Scale (0–100, higher scores better).
§FES-I: Falls Efficacy Scale-International (short version): scores range from 7 (no concern about falling) to a maximum of 28 (severe concern about falling).
¶EQ-5D-5L utility, −0.594, indicating the worst possible health state, to 1.0 and is anchored at 0 (death) and 1.0 full health.
**EQ-VAS gives self-rated health on a scale where the endpoints are labelled from 'worst imaginable health state' (0) to 'best imaginable health state' (100).
††VAS: Visual Analogue Scale, 0–100 scale, higher scores worse.
‡‡Excluding index ankle fracture.

and diary. Exercise progression was based on evidence-based guidelines[21] but individualised by progressing and regressing the volume and load in line with each participant's capabilities and preferences.

The progressive exercise intervention used simple health behaviour change techniques to optimise adherence to home exercise.[22] Participants were asked to identify their goals following usual physiotherapy practice and, with the treating therapist's help, write an action plan for where and when they would perform their home exercises and a contingency plan for managing difficulties. Therapists were trained to focus on helping participants identify barriers to exercise and becoming more physically active post injury, and facilitating problem-solving. A high-quality information booklet was developed by the AFTER study team with patient and public involvement representatives and provided to participants.

### Training and monitoring of intervention delivery

Treating therapists were trained in a face-to-face session of up to 4 hours and provided with written materials on the theory and practical delivery of the interventions. The same therapists were trained in both interventions by the trial research physiotherapist (an experienced clinician with specialist postgraduate musculoskeletal specialist training, who was also the chief investigator for the study). Intervention delivery was recorded in logs for each contact with participants to monitor intervention fidelity and enable ongoing feedback to manage contamination. Local site principal investigators were also responsible for monitoring intervention delivery.

Best practice advice delivery adherence was defined as attendance of at least one physiotherapy session. For progressive exercise delivery, adherence was defined as attending all six sessions, or if they were discharged by the physiotherapist as treatment completed (as marked on treatment log), or discharged by the physiotherapist following a patient-initiated follow-up period with no further contact, or if they were referred on for further investigations/treatment. Partial adherence for progressive exercise was attending at least one session but not meeting the outlined criteria for full adherence.

Additional sessions were recorded for any best practice advice sessions after the main first sessions and for progressive exercise it as any sessions in addition to the six sessions. For the progressive exercise programme, treatment logs captured exercise prescription details so that progression of the programme over the sessions in terms of exercise type, volume or load could be assessed.

### Concomitant care

Other aspects of health and social care continued as normal. Additional treatments, including contact with their general practitioner or other health professionals, were recorded in participant follow-up questionnaires.

### Outcome measures

#### Feasibility success criteria

The main aim of this pilot RCT was to determine the feasibility of a future definitive trial.[23] The main uncertainty was whether patients find it acceptable to be randomised to different types of rehabilitation provision. To determine the feasibility of a definitive RCT, the prespecified success criteria[17] were:

► A study participation rate of at least 25% of those eligible.
► At least 48 eligible patients across at least three sites agree to participate over a maximum of 18 months.
► At least 85% of participants completed study intervention sessions.
► At least 80% of participants attended study follow-up at 6 months, defined as the proportion of participants who provided Olerud and Molander Ankle Score (OMAS) scores at 6 months.

### Outcomes

Participants were followed-up 3 months after randomisation with a postal questionnaire and 6 months after randomisation face-to-face at the hospital. They were offered telephone or postal follow-up if they were unable to attend the 6-month follow-up.

Patient-reported outcomes at 3 and 6 months were:

► Ankle-related symptoms and function: OMAS (0–100, higher scores better).[24]

**Table 3** Compliance with the allocated treatment by trial arm

|  | Best practice advice | Progressive exercise |
|---|---|---|
| Totals | N=30 | N=31 |
| Completed exercise treatment*, n (%) | 28 (93.3) | 18 (58.1) |
| Partial exercise completion†, n (%) | 28 (93.3) | 28 (90.3) |
| Received no treatment, n (%) | 2 (6.7) | 3 (7.1) |
| DNA/unable to contact | 0 (0) | 1 (3.2) |
| Withdrawal/declined | 0 (0) | 0 (0) |
| Too difficult to attend appointments | 0 (0) | 1 (3.2) |
| Unknown | 2 (6.7) | 1 (3.2) |
| Median number of sessions (IQR) | 1 (1–2) | 5 (3–6) |
| One sessions, n | 20 | 3 |
| Two sessions, n | 5 | 2 |
| Three sessions, n | – | 3 |
| Four sessions, n | 2 | 5 |
| Five sessions, n | 1 | 7 |
| Six sessions, n | – | 7 |
| Seven sessions, n | – | 1 |
| Participants receiving additional sessions ‡, n (%) | 8 (26.7) | 1 (3.2) |
| Before protocol change § | 3 | 1 |
| After protocol change § | 5 | 0 |
| Total number of additional contact sessions, n | 13 | 1 |
| Telephone | 2 | 0 |
| Face-to-face | 11 | 1 |

*Best practice advice: attendance at ≥1 session; progressive exercise: six sessions attended, or discharged by clinician as treatment completed (as marked on treatment log), or discharged by clinician following patient-initiated follow-up period with no further contact, or referred on for further investigation/treatment.
†Defined as attendance at ≥1 session.
‡Additional sessions defined as >1 for best practice advice, and >6 for progressive exercise.
§All the extra sessions for the best practice advice arm before the protocol change were face-to-face, of the five participants who got additional sessions after the protocol change, two of them were by telephone and three were face-to-face.

- ► Lower-limb function limitations: Lower Extremity Functional Scale (0–100, higher scores better).[25]
- ► Pain: Visual Analogue Scale (VAS), 0–100 scale, higher scores worse.
- ► Health-related quality of life: EuroQol EQ-5D-5L score.[26] The EQ-5D health status scale is mapped onto the EQ-5D-3L valuation set using the Crosswalk Index Value Calculator.[27] The scale from this set ranges from −0.594, indicating the worst possible health state, to 1.0 and is anchored at 0 (death) and 1.0 (full health). A respondent's EQ-VAS gives self-rated health on a scale where the endpoints are labelled from 'worst imaginable health state' (0) to 'best imaginable health state' (100).
- ► Fear of falls: Falls Efficacy Scale-International (short version).[28] The overall scores range from 7 (no concern about falling) to a maximum of 28 (severe concern about falling).

- ► Self-efficacy: self-efficacy exercise score.[29] Overall scores range from 0 to 90, with higher scores indicating greater confidence to exercise.
- ► Self-reported return to desired activities, including work, social life and sport activities.
- ► Walking aid use and distances.
- ► Exercise adherence (self-reported exercise frequency).
- ► Health resource use (including additional out of trial physiotherapy).

At 6-month follow-up, a blinded outcome assessor collected objective measures of ankle function and physical performance:

- ► Ankle joint range: hand-held goniometry.[30]
- ► Muscle strength: hand-held dynamometry (Lafayette Manual Muscle Test System, Lafayette Instrument, Indiana, USA) of ankle dorsi/plantar flexion using a 'make' approach (working up to a maximal contraction over a maximum of 5 s and without pushing into pain and the assessor maintaining position of the device).[31] Participants were measured three times and had at least 10 s rest between attempts.
- ► Mobility and balance: short physical performance battery (SPPB).[32] The test involves physical tests of balance, walking speed and repeated rises from a chair.

### Adverse events

Expected general side effects of any exercise, such as delayed-onset muscle soreness and temporary increases in pain of less than a week, were not recorded as adverse events. Pain increases of more than a week were recorded in patient-reported questionnaires. Any exacerbations of other medical conditions during exercise or exercise-related injurious falls could also be recorded in patient-reported questionnaires, or by the site investigators if they became aware of such an event.

### Sample size

The main feasibility objective and therefore the basis of the sample size estimate was participant recruitment per centre. The target sample size was a minimum of 48 participants in at least three centres over a maximum of 18 months. The sample size was based on a target of recruiting 1.5 participants per month per site. An amendment to the protocol during recruitment enabled the recruitment to continue up to 60 participants so that more feasibility data could be collected after the changes to the best practice advice intervention outlined above.

### Statistical analysis

Baseline characteristics and outcomes were reported using descriptive statistics using mean and SD (or median and IQR if non-normally distributed), and minimum and maximum, for continuous variables and number and percentage of participants in each group for binary or categorical variables. Feasibility outcomes were reported as numbers and percentages and compared with the

**Table 4** Intervention delivery

| Treatment components | Best practice advice only, N (received BPA)=28 | Progressive exercise, N (received at least one session)=28 |
|---|---|---|
| Exercises prescribed (N,%) | 27 (96.4) | 23 (82.1) |
| Information booklet provided (N,%) | 28 (100) | 24 (85.7) |
| Session time in minutes (median, IQR) | 45 (40–60) | First session: 45 (45–40)<br>Follow-up sessions: 30 (30–45) |

BPA, Best Practice Advice.

progression criteria targets. Outcome data were analysed on an intention-to-treat basis, with all participants analysed as per their treatment group allocation and differences in outcomes between treatment groups were reported with 95% CIs. As the study was not powered for formal hypothesis testing between the treatment arms no p values are provided.

We used the Consolidated Standards of Reporting Trials (CONSORT) guidelines for pilot and feasibility trials[33][34] and the Template for Intervention Description and Replication (TIDieR) statement to report the study.[35]

## Patient and public involvement

The intervention development and study materials were supported by a patient and public involvement group, who were also involved in interpreting the study findings and definitive trial protocol design.

## RESULTS

### Screening, recruitment and baseline characteristics

Patient enrolment began in October 2018 and was completed in August 2019 when the sample size was reached. Of the 226 patients screened, 112 were eligible, and 61 were recruited and randomised (see figure 1). The main reason screened patients were not eligible was that they were unable to attend outpatient physiotherapy at a participating centre (41/114, 36%). Of the 34 patients who declined participation, the main reason provided was that they did not want to return to the participating centre for the study due to practical difficulties such as transport and parking (8/34, 24%). The revised sample size target of 60 was exceeded as one participant had consented to participation and randomised before closure of the randomisation system.

Baseline characteristics are summarised in table 2. Participants had a mean age of 63.5 (SD 8.9) years, and 42/61 (69%) were women. Also, 41/61 (67%) of the participants had surgery and the remainder (20/61, 33%) had non-surgical treatment for their ankle fracture.

### Feasibility outcomes

#### Patient engagement with the trial

The final recruitment rate of eligible participants was 54%. The overall recruitment rate per month per site was 1.4 participants.

### Intervention adherence levels

The mean time from randomisation to having the first physiotherapy session was 39.2 days (95% CI 27.3 to 55.3) for best practice advice and 38.2 (95% CI 29.8 to 48.1) for progressive exercise. From injury to starting physiotherapy it was 74.1 days (95% CI 53.9 to 94.1 days) for best practice advice and 72.7 days (95% CI 54.7 to 88.9) for progressive exercise.

At least one intervention session was delivered in 28/30 (93%) best practice advice and 28/31 (90%) progressive exercise participants. 18/31 (58%) participants met the criteria for fully completing the progressive exercise intervention (see table 3). Overall, the levels of fidelity in delivery of the intervention components were high (see table 4). The median number of physiotherapy sessions for best practice advice was 1 (IQR 1–2) and for progressive exercise it was 5 (IQR 3–6).

For the progressive exercise programme, there was evidence in treatment logs of exercise prescription progression (exercise type, volume or load) across sessions for 21/28 (75%) of participants. Exercises were not progressed for one participant, and none regressed over the sessions. It was not possible to assess for six participants due to insufficient exercise prescription data.

At 3 months post randomisation, the proportion of participants performing home exercises out of those providing data in the best practice advice and progressive exercise groups was 20/23 (87%) and 21/25 (84%), respectively. At 6 months this reduced to 14/26 (54%) in the best practice advice group and 19/26 (73%) in the progressive exercise group.

Of the 28 participants allocated to best practice advice group that received at least the first session, overall, 13 (46%) had further contacts with their physiotherapist. Eight (29%) had further contacts with their physiotherapist prior to the protocol change (see Methods). Four participants that had an additional physiotherapy session (4/28, 14%) had clinical reasons for having a further face-to-face session (no more than two sessions provided) due to issues with pain and concerns cited regarding progress and discolouration of the ankle. The other four participants had issues with appointment timing and scheduling, two participants needed extra face-to-face sessions as they had their first sessions scheduled too early (before they were permitted to weight bear), another participant arrived too late for their appointment so had to come back to complete delivery and one participant

**Table 5** Summary statistics for each clinical outcome score

| | Best practice advice | | | Progressive exercise | | | Mean difference (95% CIs) | |
|---|---|---|---|---|---|---|---|---|
| | n | Mean | SD | n | Mean | SD | Mean difference | 95% CIs |
| **OMAS*** | | | | | | | | |
| Baseline | 29 | 31.38 | 17.97 | 30 | 21.50 | 15.71 | | |
| Month 3 | 28 | 65.54 | 20.29 | 26 | 62.50 | 23.59 | | |
| Month 6 | 28 | 73.39 | 22.07 | 26 | 77.12 | 21.73 | 1.49 | (−6.87 to 9.86) |
| Change from baseline | 27 | 43.70 | 14.58 | 25 | 55.00 | 24.87 | −11.30 | (−22.55 to 0.04) |
| **LEFS†** | | | | | | | | |
| Baseline | 28 | 25.07 | 13.39 | 31 | 17.52 | 11.68 | | |
| Month 3 | 26 | 54.77 | 12.85 | 26 | 57.38 | 17.41 | | |
| Month 6 | 25 | 65.36 | 11.45 | 27 | 61.52 | 20.05 | 1.02 | (−6.28 to 8.32) |
| Change from baseline | 23 | 40.57 | 12.78 | 27 | 44.59 | 20.17 | −4.03 | (−13.83 to 5.77) |
| **FES-I‡** | | | | | | | | |
| Baseline | 28 | 14.75 | 5.18 | 27 | 17.00 | 6.16 | | |
| Month 3 | 25 | 10.04 | 3.22 | 26 | 9.65 | 3.72 | | |
| Month 6 | 26 | 9.35 | 3.02 | 26 | 9.81 | 5.02 | −0.62 | (−2.08 to 0.83) |
| Change from baseline | 26 | −5.31 | 5.12 | 23 | −8.30 | 5.34 | 3.00 | (−0.011 to 6.00) |
| **EQ-5D utility§** | | | | | | | | |
| Baseline | 29 | 0.48 | 0.29 | 30 | 0.25 | 0.30 | | |
| Month 3 | 27 | 0.72 | 0.20 | 25 | 0.74 | 0.14 | | |
| Month 6 | 28 | 0.77 | 0.15 | 28 | 0.81 | 0.27 | −0.014 | (−0.108 to 0.080) |
| Change from baseline | 27 | 0.30 | 0.31 | 27 | 0.57 | 0.35 | −0.27 | (−0.45 to to 0.09) |
| **EQ-5D VAS¶** | | | | | | | | |
| Baseline | 29 | 69.52 | 19.11 | 30 | 68.63 | 20.04 | | |
| Month 3 | 27 | 74.37 | 22.93 | 26 | 83.27 | 15.53 | | |
| Month 6 | 28 | 82.07 | 17.85 | 28 | 81.36 | 18.21 | 2.03 | (−3.53 to 7.58) |
| Change from baseline | 28 | 11.56 | 15.01 | 28 | 14.04 | 21.71 | 1.47 | (−4.26 to 7.20) |
| **SPPB**** | | | | | | | | |
| Month 6 | 22 | 10.27 | 1.91 | 25 | 10.24 | 1.59 | 0.12 | (−0.82 to 1.05) |
| Month 6†† | 27 | 8.48 | 4.41 | 28 | 9.36 | 3.60 | 0.98 | (−1.24 to 3.20) |
| **Range of motion (RoM) at 6 months (°)** | | | | | | | | |
| Dorsiflexion (uninjured) | 24 | 10.8 | 5.4 | 26 | 12.5 | 8.7 | | |
| Dorsiflexion (injured) | 24 | 9.2 | 5.1 | 26 | 7.6 | 9.0 | | |
| Dorsiflexion (difference) | 24 | 1.7 | 3.3 | 26 | 4.9 | 8.1 | | |
| % RoM dorsal flexion‡‡ | | **85.2** | | | **60.8** | | **3.55** | **(−1.03 to 7.20)** |
| Plantar flexion (uninjured) | 24 | 47.8 | 17.0 | 26 | 43.5 | 14.7 | | |
| Plantar flexion (injured) | 24 | 43.0 | 16.2 | 26 | 38.0 | 12.2 | | |
| Plantar flexion (difference) | 24 | 4.8 | 10.1 | 26 | 5.5 | 7.0 | | |
| % RoM plantar flexion‡‡ | | **90.0** | | | **87.4** | | **0.22** | **(−4.71 to 5.14)** |
| Inversion (uninjured) | 24 | 33.5 | 10.1 | 26 | 31.5 | 10.8 | | |
| Inversion (injured) | 24 | 26.0 | 10.7 | 26 | 26.3 | 11.7 | | |
| Inversion (difference) | 24 | 7.5 | 8.0 | 26 | 5.2 | 8.7 | | |
| % RoM inversion‡‡ | | **77.6** | | | **83.5** | | **−3.19** | **(−7.49 to 1.10)** |
| Eversion (uninjured) | 23 | 18.5 | 6.6 | 26 | 17.1 | 6.3 | | |
| Eversion (injured) | 23 | 15.0 | 5.6 | 26 | 15.6 | 7.5 | | |
| Eversion (difference) | 23 | 3.5 | 5.5 | 26 | 1.5 | 6.3 | | |
| % RoM eversion‡‡ | | **81.2** | | | **91.2** | | **−1.70** | **(−5.24 to 1.84)** |
| **Strength§§ at 6 months (Newtons)** | | | | | | | | |
| Dorsiflexion strength | 24 | 151.3 | 54.4 | 26 | 174.2 | 99.9 | 32.24 | (2.85 to 61.64) |
| Plantar flexion strength | 24 | 221.1 | 85.2 | 26 | 212.4 | 107.8 | 6.34 | (−28.83 to 41.50) |
| **Pain at rest VAS¶¶** | | | | | | | | |

Continued

**Table 5** Continued

| | Best practice advice | | | Progressive exercise | | | Mean difference (95% CIs) | |
|---|---|---|---|---|---|---|---|---|
| | n | Mean | SD | n | Mean | SD | Mean difference | 95% CIs |
| Baseline | 29 | 26.6 | 28.2 | 31 | 32.1 | 27.5 | | |
| Month 3 | 26 | 11.6 | 18.1 | 26 | 16.8 | 24.0 | | |
| Month 6 | 25 | 6.5 | 13.6 | 28 | 8.2 | 22.7 | −0.82 | (−6.64 to 5.01) |
| Pain walking VAS¶¶ | | | | | | | | |
| Baseline | 28 | 37.8 | 30.0 | 26 | 55.2 | 27.0 | | |
| Month 3 | 27 | 19.1 | 23.5 | 25 | 19.6 | 24.1 | | |
| Month 6 | 26 | 12.2 | 19.6 | 27 | 10.7 | 22.8 | −1.11 | (−9.61 to 7.40) |
| Self-Efficacy for Exercise Scale*** | | | | | | | | |
| Month 3 | 18 | 61.7 | 27.2 | 20 | 71.0 | 22.0 | | |
| Month 6 | 23 | 69.6 | 21.2 | 25 | 67.8 | 21.5 | 2.24 | (−12.11 to 16.60) |

*OMAS: Olerud and Molander Ankle Score (0–100, higher scores better).
†LEFS: Lower Extremity Functional Scale (0–100, higher scores better).
‡FES-I: Falls Efficacy Scale-International (short version), scores range from 7 (no concern about falling) to a maximum of 28 (severe concern about falling).
§EQ-5D-5L utility, −0.594, indicating the worst possible health state, to 1.0 and is anchored at 0 (death) and 1.0 full health.
¶EQ-VAS gives self-rated health on a scale where the endpoints are labelled from 'worst imaginable health state' (0) to 'best imaginable health state' (100).
**SPPB: short physical performance battery (scores range from 0 to 12, higher scores better).
††Imputes 0 for those participants who completed form, but left questions blank—they were assumed to have not attempted.
‡‡Calculated using the injured divided by the uninjured scores, multiplied by 100 to output as a percentage.
§§The highest strength score (measured in Newtons N) out of 3 was used for each patient.
¶¶VAS: Visual Analogue Scale (0–100 scale, higher scores worse).
***Self-Efficacy Exercise Score. Overall scores range 0–90, with higher scores indicating greater confidence to exercise.

was initially seen in physiotherapy in error and had a usual care physiotherapy appointment, they had an extra session to receive the best practice advice session. Of the five participants who had additional contact with their physiotherapist after the protocol change, two were by telephone and three were face-to-face. One participant in the progressive exercise group had a further contact in addition to the six physiotherapy sessions. Three participants in each group reported receiving sessions of physiotherapy outside the trial (see health resource use data in the online supplemental tables S1 and S2).

### Patient retention and outcome measure data collection
The patient-reported outcomes and objective measures of ankle function and physical performance are reported in table 5. Follow-up rate for OMAS scores at 6 months was 28/30 (93%) for the best practice advice group and 26/31 (84%) in the progressive exercise group, an overall follow-up rate of 89%. Completion rates for the

objective outcome assessments at 6 months were lower than for the OMAS (eg, 47/61 (77%) for SPPB). Data on return to desired activities, including work, social life, and sport activities, walking aid use and distances, and health resource use are reported in the supplementary appendix (see online supplemental tables S1 to S4).

### Adverse events
No serious adverse events were reported. Increases in pain of more than 1 week were reported by three participants in the best practice advice group and two in the progressive exercise group, all at the 3-month time point. One participant in the progressive exercise group consulted a physician due to this symptom increase. No increases in pain were reported at 6 months. Five participants reported a worsening of a medical condition at 3 months (three in the best practice advice group and two in the progressive exercise group). Of these reports, 3/5 (60%) sought medical attention as a result. One participant in

**Table 6** Feasibility assessment

| | Target | Outcome |
|---|---|---|
| Participation rate | A study participation rate of at least 25% of those eligible, to indicate acceptability and generalisability. | Sixty-one of 112 (54%) eligible participants participated in the trial, exceeding the feasibility criteria for participation of 25%. |
| Recruitment | At least 48 eligible participants across at least three sites agreed to participate over a maximum of 18 months. | Sixty-one participants were recruited in 11 months at five centres. There were more sites than originally planned so to interpret this feasibility criteria the recruitment rate per centre month was more appropriate. The recruitment feasibility criteria were based on 1.5 participants per site per month. 1.4 participants per centre month was achieved during the pilot trial. |
| Completion of study intervention sessions | At least 85% of participants completed study intervention sessions. | At least one intervention was provided for 28/30 (93%) best practice advice and 28/31 (90%) progressive exercise participants. 18/31 (58%) participants met the criteria for fully completing the progressive exercise intervention. |
| Follow-up rate | At least 80% of participants attended study follow-up at 6 months. | Defined as the number who provided Olerud and Molander Ankle Score scores at 6 months, was 28/30 (93%) for the best practice advice group and 26/31 (84%) in the progressive exercise group, an overall follow-up rate of 89%. |

the best practice group reported a worsening of a medical condition at 6 months and saw a physician. Adverse events are reported in the supplementary appendix online supplemental table S5.

A summary of the prespecified feasibility success criteria and the respective outcomes are shown in table 6.

## DISCUSSION

Our pilot RCT demonstrated that a definitive trial comparing a best practice advice versus a supervised progressive exercise intervention for adults aged 50 years and over after ankle fracture is feasible. All feasibility success criteria were met, apart from a minor difference in the recruitment rate per site-month that would need to be factored into planning a definitive trial. Additionally, ensuring similar rates of follow-up in each intervention group is critical.

Main areas of refinement for the definitive RCT indicated by the pilot related to the interventions. Most patients had to wait several weeks after removal of the cast/splint for their first physiotherapy session. It would therefore be ideal for the early rehabilitation advice to start at the point where the cast or splint are removed rather than having to wait for a separate outpatient physiotherapy appointment. The most common reason participants did not want to participate was the requirement to attend the hospital for appointments. With the now widespread uptake of videoconferencing to deliver physiotherapy in the UK, a future trial where this is available as an option could make the trial more inclusive.

In the AFTER pilot RCT we observed that in the best practice advice group about two in three participants had a single session of advice and did not use the optional additional sessions. Half of the additional sessions had clinical reasons, the remainder had issues with appointment timing and scheduling. The clinical reasons provided for the additional session seemed to relate to concerns regarding progress and symptoms that were concerning but anticipated after ankle fracture. The advice intervention could be improved with further information on symptoms and rate of progress to help address these early concerns. The majority of participants managed with a single session of advice without initiating the optional extra sessions. There was also minimal and balanced use of physiotherapy outside the trial over the 6 months (three participants in each intervention group, 6/61 (10%) in total). This contrasts to the previously largest trial of advice versus supervised rehabilitation, the EXACT trial that was based in Australia,[15] where 39/108 (36%) of the advice group and 15/106 (14%) of the physiotherapy supervised rehabilitation group had additional physiotherapy outside the trial, 54/214 (25%) in total.

In comparison to the outcome measure completion rates for patient-reported outcomes, there were lower rates of objective measures of ankle function and physical performance. Given the greater extent of missing outcome data and the extra clinical and participant burden, and the context of trying to reduce hospital visits since the start of the COVID-19 pandemic, use of remote follow-up with questionnaires only is an important option for the definitive RCT.

We asked participants to self-report adverse events as one of the interventions had a much greater level of contact with a health professional and it was anticipated that there would be few serious adverse events. However, the validity of the relatedness of adverse events to the participants rehabilitation is difficult to assess when dependant on self-report. The recruiting centres could also report adverse events, this aimed to identify the more serious treatment-related complications. The progressive exercise programme had a higher intensity than the best practice advice programme. The higher intensity appeared to be generally well tolerated as there were few adverse events, no serious adverse events and those events reported were similar between the intervention groups.

A limitation of the study, common to all pilot trials is that with the necessarily limited number of centres, there remains some uncertainty regarding the feasibility of a trial across a much larger number of centres. We attempted to mitigate this issue by running the trial at a range of trauma units and major trauma centres. These centres had varied geographical settings, diverse populations and a range of levels of experience in running multicentre trauma trials. Physiotherapists were trained in both interventions so there was a potential risk of contamination. Although treatment logs indicated good levels of intervention fidelity, more on-site observation and/or recording of interventions would have enabled a more robust assessment of fidelity. Training separate physiotherapists in each intervention would also be advisable where it is practical to do so to further reduce the risk of contamination.

The evidence from this pilot RCT demonstrated that with refinement, a definitive trial would be feasible. A definitive trial was designed based on the findings from the external pilot and has subsequently been funded (National Institute for Health and Care Research reference: NIHR201950). The definitive AFTER trial will assess the clinical effectiveness of physiotherapist-supervised rehabilitation versus self-directed rehabilitation for adults aged 50 years and over after ankle fracture. The interventions, study process and outcome assessments have all been refined for the full trial. The primary endpoint will be the OMAS at 6 months. The full AFTER is due to complete in July 2024.

**Author affiliations**
[1]Oxford Trauma and Emergency Care, Nuffield Department of Orthopaedics, Rheumatology and Musculoskeletal Sciences, University of Oxford, Oxford, UK
[2]Faculty of Health and Life Sciences, University of Exeter, Exeter, UK
[3]Oxford Clinical Trials Research Unit, University of Oxford, Oxford, UK

4Kadoorie Research Centre, Oxford University Hospitals NHS Foundation Trust, Oxford, UK
5Leeds Institute of Rheumatology and Musculoskeletal Medicine, University of Leeds, Leeds, UK

**Acknowledgements** This study was conducted as part of the portfolio of trials in the registered UKCRC Oxford Clinical Trials Research Unit (OCTRU) at the University of Oxford. It followed their Standard Operating Procedures, ensuring compliance with the principles of Good Clinical Practice and the Declaration of Helsinki and any applicable regulatory requirements. Special thanks to Nicola Kenealy, Molly Glaze, Gemma Greenall, Rebecca MacLean, Emma Haines, Lucy Eldridge, Juul Achten, Kylea Draper, Elisa Basso, Amrita Athwal and the wider team at the OCTRU and all the physiotherapists, surgeons and research associates at the recruiting centres. We also wish to thank attendees of an intervention development meeting, including: Ann Tomline, Rebecca Hibbs, Deb Smith, Elizabeth Houghton, Gareth Boyden, Georgina Taylor, Jacqueline Claydon, Jon Room, Jonathan Young, Jean Millar, Julie Wright, Karen Keates, Kate Bennett, Katherine Coates, Katie Sheehan, Liz Baird, Mark Williams, Philip Bell, Pippa Ellery, Richard Grant, Sian MacRae, Suzanne Jones, Thavapriya Sugavanam, Trisha Richardson. Thanks also to Colin Forde for contributing to the intervention development process.

**Collaborators** The AFTER study collaborating principal investigators were: Hannah Perkins (Oxford University Hospitals NHS Foundation Trust), Carol McCrum (East Sussex Healthcare NHS Trust), Jacky Jones (Guy's and St Thomas' NHS Foundation Trust), Carey McClellan and Naomi Chalk (University Hospitals NHS Foundation Trust) and Fiona Cowell (Royal Liverpool and Broadgreen University Hospital NHS Trust).

**Contributors** DK was the chief investigator and is the guarantor, he conceived and designed the study, was awarded the funding and drafted the manuscript. MLC, ET, SH, VSB, SD, AR, KW and SL contributed to study design and provided specific content and edited the manuscript. SD oversaw the statistical aspects of the study. VSB provided trial management oversight. All authors have reviewed and approved the manuscript.

**Funding** This report is independent research supported by the National Institute for Health and Care Research (NIHR Post-Doctoral Fellowship, DK, PDF-2016-09-056). The report was supported by the NIHR Biomedical Research Centre, Oxford. SL received funding from the NIHR Collaboration for Leadership in Applied Health Research and Care Oxford at Oxford Health NHS Foundation Trust.

**Competing interests** The institutions of the authors have received research grant funding from National Institute for Health Research, European Union, Royal College of Surgeons England and industry.

**Patient and public involvement** Patients and/or the public were involved in the design, or conduct, or reporting, or dissemination plans of this research. Refer to the Methods section for further details.

**Patient consent for publication** Not applicable.

**Ethics approval** This study involves human participants and was approved by Hampshire B Research Ethics Committee (18/SC/0281), approval 2 July 2018. Participants gave informed consent to participate in the study before taking part.

**Provenance and peer review** Not commissioned; externally peer reviewed.

**Data availability statement** Data are available upon reasonable request. Data sharing requests can be considered via contact with the corresponding author.

**ORCID iDs**
David J Keene http://orcid.org/0000-0001-7249-6496
Elizabeth Tutton http://orcid.org/0000-0003-3973-360X

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
