## [Reviewer comments · BMJ Open]

ARTICLE DETAILS

TITLE (PROVISIONAL)	Progressive exercise versus best practice advice for adults aged 50 years or over after ankle fracture: the AFTER pilot randomised controlled trial
AUTHORS	Keene, David; Costa, Matthew; Peckham, Nicholas; Tutton, Elizabeth; Barber, Vicki; Dutton, Susan; Hopewell, Sally; Redmond, Anthony; Willett, Keith; Lamb, Sarah; trial collaborators, AFTER

VERSION 1 – REVIEW

REVIEWER	Rodriguez, Aryelly University of Edinburgh, (ECTU) Edinburgh Clinical Trials Unit
REVIEW RETURNED	05-Jan-2022

GENERAL COMMENTS	1.- page1 line 8, please update "The aim of the AFTER study, an external multicentre pilot randomised controlled trial, was" to "The aim of the Ankle Fracture Treatment Enhancing Rehabilitation (AFTER) study, an multicentre pilot parallel-group randomised controlled trial (RCT), was". 2.- page 1 line 46, you wrote "were fully or partially delivered ", since this is the abstract it would be more adequate to refer to "received at least one session". 3.- page 1 line 59, please define OMAS acronym. 4.- page 5 line 11, you wrote "at 16 per 10,000 person-years in women aged 60 to 70", I have been to this reference and I cannot find this, please double check. 5.- page 5, line 22, please define AIM acronym. 6.- page 7, line 55, you wrote "Where possible", where there instances in which it was not possible? I would recommend you remove those words. Later on the text (page 11 line 53) you are more precise and wrote "a blinded outcome assessor collected.." 7.- page 17, line 11, please update "physical outcome measures" to "objective measures of ankle function and physical performance" 8.- page 17, line 29, you wrote "Overall, the challenging progressive exercise", his is the first time that the intervention is described a challenging, do you mean that it is "more physically demanding" for patients, or more time consuming or more difficult to deliver? I think you just need a little bit of precision in the wording. 9.- page 17, line 60, you wrote "and has subsequently been funded.", Congratulations on my behalf. Please, if feasible reference your grant number. 10.- Table 2, please, if feasible, update with footnotes and as indicated in the attached pdf file. 11.-As per the protocol paper please define the intention to treat population somewhere in the results 12.- An excellent manuscript, I have multiple very minor comments regarding spelling, formats, missing text (including text and tables), These can be seen in the attached pdf copy of your manuscript with my comments in red letters.
---

REVIEWER	Sheehan , Katie Kings College London, Academic Department of Physiotherapy, Division of Health and Social Care Research I currently sit on the Fragility Fracture Network Scientific Committee which Costa (co-author) is a member of. I have published with, and developed grant proposals with Lamb (co-author).
-----------------	---

	I have received funding from NIHR (funder) for fragility fracture research not related to the current study
REVIEW RETURNED	28-Jan-2022

GENERAL COMMENTS	Thank you for the opportunity to review this manuscript which sought to determine the feasibility of a trial comparing rehabilitation approaches for adults aged 50 years and over after an ankle fracture. Overall, this is a well written report of the trial. My comments are mostly to improve transparency and consistency for the reader. I look forward to the results of the definitive trial. Introduction: Paragraph 1: Suggest removing 'are very common'. Paragraph 2: Reference for 'function is poor due to pain' Objectives: For patient engagement and retention 'measured by' is provided. This is not provided for adherence levels and/or acceptability. It would be good to include for consistency. Methods: Blinding - Was the intention that all would be blind but that in some cases this was not achieved as suggested by the 'where possible' at the start of the sentence? This does not seem to be the case as later indicate ' at 6-month follow-up, a blinded outcome assessor' – would be beneficial to clarify. Intervention – for therapist training indication that was up to 4 hours for both interventions. In Table 1 suggests the training were distinct (and half day +2-3 hours). Would be good to ensure consistency. Intervention - Table 1 for 'when and how much' for the practice advice group it would be good to specify (vs. 'volume of exercise detailed in the manuscript') so the table can be reviewed as a stand-alone summary of the intervention. Similar comment for 'how well:reporting'. Monitoring of intervention delivery – no mention of monitoring for progression of exercise prescription but this later is outlined in the results. Would be good to add here. Page 9 'feasibility success criteria' – could be condensed as repeats the aim stated at the end of the introduction. Would be good to indicated these were specified a priori and cite the protocol. Adverse events – 'become' should read 'became' Statistical analysis –it would be transparent to include description of analysis for feasibility outcomes which differ from baseline characteristics and outcomes – e.g., with 95% confidence intervals. Table 2: for 'falls that resulted in a broken bone in last 3 months' should they not all be yes? or there was different mechanisms of injury for the ankle fractures? maybe I've missed something here! For compliance with treatment arm the text (and Table 5) are perhaps a little misleading given for the progressive exercise group 58.1% completed the exercise treatment (vs. 93.3% for the best practice advice) as specified in Table 3. May be beneficial to cite for completed and partial separately. For additional sessions the paragraph should perhaps start with the total who received additional contact (n = 13) vs. the 8 before the protocol change (and the 5 after later specified) to avoid confusion when comparing against table 3 (I had to reread a few times). I did not see a supplementary appendix in the submission so could not review data on return to desired activities, including work, social life, and sport activities, walking aid use and distances, and health resource use. Table 6 – not clear when ROM/strength were measured. Table 6 - It was a pity not to see missing data explicitly specified in Table
---

	6/outlined in more detail in text. Particularly given the discussion indicated that physical measures had greater missing data than patient-reported outcome measures. Granted, for a definitive trial, questionnaires will make life easier, but the extent of missing data in the pilot is perhaps not a convincing justification (47 participants at 6-months on the SPPB does not seem that much worse than some of the patient reported outcome measures ~52 completed at 6-months). Discussion – paragraph 2 line 1 remove ‘the’ Discussion – is the difference in adherence between AFTER and EXACT trials due to the nature of the trial or the nature of the setting (EXACT is in Australia where access to physiotherapy may be more readily available)? Discussion/conclusion: I was left wondering about the definitive trial. Would it be possible to conclude with the research question for this trial (at no point do you indicate the primary outcome for the definitive trial), when the trial is due to be completed by, and perhaps a reference to the trial registration so that people could look it up? As it stands the concluding sentence related to value for money from the funders perspective seems out of place and a bit generic. Additional: Throughout -switch between ‘participants’ and ‘patients’ – suggest one terms for consistency.
--	--

REVIEWER	Maddocks, Stacy University of KwaZulu-Natal, Physiotherapy, rehabilitation services for children
REVIEW RETURNED	08-Jun-2022

GENERAL COMMENTS	Thank you for your submission. While I find your work interesting I found the presentation of this paper in need of a few improvements. Firstly the title of your study is quite misleading, " Progressive exercise versus best practice advice for adults aged 50 years or over after ankle fracture: the AFTER pilot randomised controlled trial"- the word feasibility does not appear in the title yet your overall objective is to determine the feasibility of this intervention. The title sounds like a therapeutic comparison of interventions not a feasibility study. The style of writing was too concise and poorly explanatory in some aspects of the paper. Write full, complete sentences for better scientific delivery and comprehension of your message. The way the methods section is structured is confusing. You discuss the study sample under Methods and later discuss it again under outcomes. As a reviewer I had a sense that the manuscript was rushed and or had too many different people contributing small parts that were poorly fit together. Your sample (the technique of sampling and calculating the appropriate sample size) should be discussed under Methods and your sample (participant) characteristics can later be discussed under your results as participant demographics. The same goes for the study time frame. that all needs discussion under your Methods section. Under the heading Training and monitoring...you said that the treating therapists were trained in a face-to-face session of up to four hours and provided with written materials on the theory and practical delivery of the interventions- my question to you is who provided this training and what were their credentials? What guided your definitions of your feasibility success criteria? If this paper is reconsidered and written more thoughtfully it may certainly be eligible for publication but unfortunately not at this point.
---

VERSION 1 – AUTHOR RESPONSE

Dr. Aryelly Rodriguez, University of Edinburgh

Comments to the Author:

1.- page1 line 8, please update "The aim of the AFTER study, an external multicentre pilot randomised controlled trial, was" to "The aim of the Ankle Fracture Treatment Enhancing Rehabilitation (AFTER) study, an multicentre pilot parallel-group randomised controlled trial (RCT), was".

Response 2: The text has been amended in line with this suggestion.

2.- page 1 line 46, you wrote "were fully or partially delivered ", since this is the abstract it would be more adequate to refer to "received at least one session".

Response 3: Thank you for this suggestion, we have amended the text in line with this suggestion.

3.- page 1 line 59, please define OMAS acronym.

Response 4: Now defined.

4.- page 5 line 11, you wrote "at 16 per 10,000 person-years in women aged 60 to 70", I have been to this reference and I cannot find this, please double check.

Response 5: This data was from the supplementary materials for the study where ankle specific data are provided: Online Supplementary Fig. 1. Age and sex specific incidence rates of fracture at selected sites, 1988–2012 (black points female, open points male).

5.- page 5, line 22, please define AIM acronym.

Response 6: Definition added.

6.- page 7, line 55, you wrote "Where possible", where there instances in which it was not possible? I would recommend you remove those words. Later on the text (page 11 line 53) you are more precise and wrote "a blinded outcome assessor collected.."

Response 7: Thank you for this suggestion. We have deleted 'where possible', this is in line with the protocol.

7.- page 17, line 11, please update "physical outcome measures" to "objective measures of ankle function and physical performance"

Response 8: Updated as suggested in both places in the paper it was mentioned.

8.- page 17, line 29, you wrote "Overall, the challenging progressive exercise", his is the first time that the intervention is described a challenging, do you mean that it is "more physically demanding" for patients, or more time consuming or more difficult to deliver? I think you just need a little bit of precision in the wording.

Response 9: Amended to 'The progressive exercise programme had a higher intensity than the best practice advice programme. The higher intensity appeared to be generally well tolerated...'

9.- page 17, line 60, you wrote "and has subsequently been funded.", Congratulations on my behalf. Please, if feasible reference your grant number.

Response 10: Now referenced.

10.- Table 2, please, if feasible, update with footnotes and as indicated in the attached pdf file.

Response 11: Thank you for these suggestions. We have added some of the additional information into the table directly. Some of the further information in carers was not added as we did not restrict people from indicating if they were doing paid and unpaid caring or both.

11.-As per the protocol paper please define the intention to treat population somewhere in the results

Response 12: This has been added to the statistical analysis section in the methods.

12.- An excellent manuscript, I have multiple very minor comments regarding spelling, formats, missing text (including text and tables), These can be seen in the attached pdf copy of your manuscript with my comments in red letters.

Please also see attached comments

Response 13: Thank you for this positive feedback and for the detailed and helpful suggestion to improve our manuscript, further edits have been made to the manuscript and CONSORT flow diagram, addressing the additional minor comments in the pdf.

Reviewer: 2
Dr. Katie Sheehan , Kings College London

Comments to the Author:

Thank you for the opportunity to review this manuscript which sought to determine the feasibility of a trial comparing rehabilitation approaches for adults aged 50 years and over after an ankle fracture. Overall, this is a well written report of the trial. My comments are mostly to improve transparency and consistency for the reader. I look forward to the results of the definitive trial.

Response 14: Thank you for this positive and most helpful review, we hope you agree that the manuscript has notably improved reporting following the changes in response to the points raised below.

Introduction:

Paragraph 1: Suggest removing 'are very common'.

Response 15: Removed.

Paragraph 2: Reference for 'function is poor due to pain'

Response 16: Thank you for noticing this. We have removed the comma in this sentence and added 'and' as reference number 7 is for pain and reduced range.

Objectives:

For patient engagement and retention 'measured by' is provided. This is not provided for adherence levels and/or acceptability. It would be good to include for consistency.

Response 17: Added as suggested.

Methods:

Blinding - Was the intention that all would be blind but that in some cases this was not achieved as suggested by the 'where possible' at the start of the sentence? This does not seem to be the case as later indicate ' at 6-month follow-up, a blinded outcome assessor' – would be beneficial to clarify.

Response 18: Amended as per response 7.

Intervention – for therapist training indication that was up to 4 hours for both interventions. In Table 1 suggests the training were distinct (and half day +2-3 hours). Would be good to ensure consistency.

Response 19: Thank you for spotting this error in table 1, this has been edited to ensure accurate and consistent with the text.

Intervention - Table 1 for 'when and how much' for the practice advice group it would be good to specify (vs. 'volume of exercise detailed in the manuscript') so the table can be reviewed as a stand-alone summary of the intervention. Similar comment for 'how well:reporting'.

Response 20: This comment has helped us identify some issues with Table 1 and led to amendments to both these sections of the table to ensure consistency and accuracy.

Monitoring of intervention delivery – no mention of monitoring for progression of exercise prescription but this later is outlined in the results. Would be good to add here.

Response 21: We have added 'For the progressive exercise programme, treatment logs captured exercise prescription details so that progression of the programme over the sessions in terms of exercise type, volume or load could be assessed.'

Page 9 'feasibility success criteria' – could be condensed as repeats the aim stated at the end of the introduction. Would be good to indicated these were specified a priori and cite the protocol.

Response 22: We have condensed this slightly and added that these were pre-specified and cited the protocol as suggested,

Adverse events – 'become' should read 'became'

Response 23: Amended.

Statistical analysis –it would be transparent to include description of analysis for feasibility outcomes which differ from baseline characteristics and outcomes – e.g., with 95% confidence intervals.

Response 24: Thank you we have updated the manuscript and Table 6 to include 95% CIs as specified in the protocol

Table 2: for 'falls that resulted in a broken bone in last 3 months' should they not all be yes? or there was different mechanisms of injury for the ankle fractures? maybe I've missed something here!

Response 25: A footnote has been added to Table 2 to explain the variable further and clarify 'Excluding index ankle fracture'.

For compliance with treatment arm the text (and Table 5) are perhaps a little misleading given for the progressive exercise group 58.1% completed the exercise treatment (vs. 93.3% for the best practice advice) as specified in Table 3. May be beneficial to cite for completed and partial separately.

Response 26: We have added to the text and Table 5 to ensure transparency: '18/31 (58%) participants met the criteria for fully completing the progressive exercise intervention.'

For additional sessions the paragraph should perhaps start with the total who received additional contact (n = 13) vs. the 8 before the protocol change (and the 5 after later specified) to avoid

confusion when comparing against table 3 (I had to reread a few times).

Response 27: We have restructured this section in line with the suggestion above to ensure this now reads more clearly.

I did not see a supplementary appendix in the submission so could not review data on return to desired activities, including work, social life, and sport activities, walking aid use and distances, and health resource use.

Response 28: We are not sure why this was not in the files for available review but have ensured it is in the submission files on this revision.

Table 6 – not clear when ROM/strength were measured.

Response 29: Thank you for noting this issue, timepoints have been added.

Table 6 - It was a pity not to see missing data explicitly specified in Table 6/outlined in more detail in text. Particularly given the discussion indicated that physical measures had greater missing data than patient-reported outcome measures. Granted, for a definitive trial, questionnaires will make life easier, but the extent of missing data in the pilot is perhaps not a convincing justification (47 participants at 6-months on the SPPB does not seem that much worse than some of the patient reported outcome measures ~52 completed at 6-months).

Response 30: We have carefully reviewed Table 6 and feel the extent of missing data is sufficiently specified. The number of participants providing data are summarised for each outcome. In the discussin we have highlighted that it is because the key the physical measures had a higher rate of missing data (e.g. 54/61 (89%) for the OMAS questionnaire compared to 47/61 (77%) for the SPPB), in combination the with practical challenges worsened by Covid-19, that led to the recommendation about focussing on remotely administered questionnaires for a full trial.

Discussion – paragraph 2 line 1 remove ‘the’

Response 31: removed.

Discussion – is the difference in adherence between AFTER and EXACT trials due to the nature of the trial or the nature of the setting (EXACT is in Australia where access to physiotherapy may be more readily available)?

Response 32: we have highlighted now that this was based in Australia to highlight the different context to the reader. This is an interesting point, but it is difficult to know what caused the difference as the trial populations, context and interventions all varied.

Discussion/conclusion: I was left wondering about the definitive trial. Would it be possible to conclude with the research question for this trial (at no point do you indicate the primary outcome for the definitive trial), when the trial is due to be completed by, and perhaps a reference to the trial registration so that people could look it up? As it stands the concluding sentence related to value for money from the funders perspective seems out of place and a bit generic.

Response 33: We have added some further details for the full trial as suggested. The final sentence has been removed.

Additional:

Throughout -switch between ‘participants’ and ‘patients’ – suggest one terms for consistency.

Response 34: We have reviewed the text to ensure participants is used consistently for all activities after consent and patients for pre-enrolment or out of trial contexts.

Reviewer: 3
Dr. Stacy Maddocks, University of KwaZulu-Natal

Comments to the Author:
Dear Authors

Thank you for your submission. While I find your work interesting I found the presentation of this paper in need of a few improvements. Firstly the title of your study is quite misleading, "Progressive exercise versus best practice advice for adults aged 50 years or over after ankle fracture: the AFTER pilot randomised controlled trial" - the word feasibility does not appear in the title yet your overall objective is to determine the feasibility of this intervention. The title sounds like a therapeutic comparison of interventions not a feasibility study.

Response 35: We have adopted terminology used in feasibility study frameworks, in that a randomised pilot trial is one type of feasibility study so do not consider the title misleading (see <https://journals.plos.org/plosone/article?id=10.1371/journal.pone.0150205>). A pilot trial is not powered to compare effectiveness of the interventions.

The style of writing was too concise and poorly explanatory in some aspects of the paper. Write full, complete sentences for better scientific delivery and comprehension of your message.

Response 36: While we acknowledge the manuscript had some issues, many helpfully highlighted by Reviewers 1 and 3, we do not agree that the scientific writing is broadly inadequate. We have improved the manuscript with the suggestions from Review 1 and 2 and the Editor.

The way the methods section is structured is confusing. You discuss the study sample under Methods and later discuss it again under outcomes.

Response 37: In the methods we mention the sample size, and how and why this increased during the trial. In the results we discuss the recruitment outcomes.

As a reviewer I had a sense that the manuscript was rushed and or had too many different people contributing small parts that were poorly fit together.

Response 38: We do not agree with this comment. We acknowledge that there were some errors contained in the manuscript, while this is disappointing after taking care during preparation of the manuscript, we were pleased we could rectify these with the help of Reviewer 1 and 2's suggestions.

Your sample (the technique of sampling and calculating the appropriate sample size) should be discussed under Methods and your sample (participant) characteristics can later be discussed under your results as participant demographics. The same goes for the study time frame. that all needs discussion under your Methods section.

Response 39: Our sample size is in the methods section and the recruitment and participant data is in the results section, so we are not sure how to respond to this comment. We have added 'a target of' as we think this further clarifies the details in the methods section. The study time frame should be in the results section, in accordance with the CONSORT guidelines (see item 14a here <https://www.bmj.com/content/bmj/355/bmj.i5239.full.pdf>)

Under the heading Training and monitoring...you said that the treating therapists were trained in a face-to-face session of up to four hours and provided with written materials on the theory and practical delivery of the interventions- my question to you is who provided this training and what were their credentials?

Response 40: We have added ‘...by the trial research physiotherapist (an experienced clinician with specialist post-graduate musculoskeletal specialist training, who was also the chief investigator for the study)’.

What guided your definitions of your feasibility success criteria?

Response 41: There were a priori criteria agreed by the study team, we have added some details as per response 22.

If this paper is reconsidered and written more thoughtfully it may certainly be eligible for publication but unfortunately not at this point.